# Is ROTEM Diagnostic in Trauma Care Associated with Lower Mortality Rates in Bleeding Patients?—A Retrospective Analysis of 7461 Patients Derived from the TraumaRegister DGU^®^

**DOI:** 10.3390/jcm11206150

**Published:** 2022-10-19

**Authors:** Katrin Riehl, Rolf Lefering, Marc Maegele, Michael Caspers, Filippo Migliorini, Hanno Schenker, Frank Hildebrand, Matthias Fröhlich, Arne Driessen

**Affiliations:** 1Department of Orthopaedics, Traumatology and Reconstructive Surgery, University Hospital RWTH Aachen, RWTH Aachen University, Pauwelsstrasse 30, 52074 Aachen, Germany; 2IFOM—Institute for Research in Operative Medicine, Witten/Herdecke University, Ostmerheimerstrasse 200, 51109 Cologne, Germany; 3Department for Orthopaedic Surgery, Traumatology and Sports Traumatology, Cologne Merheim Medical (CMMC), Witten/Herdecke University, 51109 Cologne, Germany; 4Department of Traumatology and Orthopaedics, Luisenhospital Aachen, Evangelischer Krankenhausverein zu Aachen von 1867, 52064 Aachen, Germany

**Keywords:** trauma haemorrhage, trauma induced coagulopathy, ROTEM, haemostatics, blood transfusion

## Abstract

Introduction: Death from uncontrolled trauma haemorrhage and subsequent trauma-induced coagulopathy (TIC) is potentially preventable. Point-of-care devices such as rotational thromboelastometry (ROTEM^®^) are advocated to detect haemostatic derangements more rapidly than conventional laboratory diagnostics. Regarding reductions in RBC transfusion, the use of ROTEM has been described as being efficient and associated with positive outcomes in several studies. Objective: The effect of ROTEM use was assessed on three different outcome variables: (i) administration of haemostatics, (ii) rate of RBC transfusions and (iii) mortality in severely injured patients. Methods and Material: A retrospective analysis of a large data set of severely injured patients collected into the TraumaRegister DGU^®^ between 2009 and 2016 was conducted. The data of 7461 patients corresponded to the inclusion criteria and were subdivided into ROTEM-using and ROTEM-non-using groups. Both groups were analysed regarding (i) administration of haemostatics, (ii) rate of RBC transfusions and (iii) mortality. Results: A lower mortality rate in ROTEM-using groups was observed (*p* = 0.043). Furthermore, more patients received haemostatic medication when ROTEM was used. In ROTEM-using groups, there was a statistically relevant higher application of massive transfusion. Conclusions: In this retrospective study, the use of ROTEM was associated with reduced mortality and an increased application of haemostatics and RBC transfusions. Prospective evidence is needed for further evidence-based recommendations.

## 1. Introduction

Coagulopathy remains a major threat to the life of severely injured patients. Next to hypothermia and acidosis, early trauma-induced coagulopathy (TIC) forms the third element of the lethal triad of trauma. Due to its association with increased morbidity and mortality [1], early and goal-directed therapy forms an essential factor in trauma care [2]. The aetiology of trauma-induced-coagulopathy is multifactorial and treatment is time-critical [3]. By using conventional laboratory diagnostics, the time between blood withdrawal and the availability of results (>80 min) [4] is exceeding the time frame when urgent therapeutic action is needed in the presence of TIC. Hence, conventional laboratory testing may not be fast enough to detect coagulopathy per se or the potential cause for it in time.

Point-of-care (POC) diagnostics such as rotational thromboelastometry (ROTEM^®^) were developed to be executed next to the patient and therefore are ready to deliver results without laborious pre-analytical sample preparation [4]. ROTEM analyses blood samples by detecting the viscoelastic properties of blood in four different modes: EXTEM for measuring the extrinsic pathway, INTEM for the intrinsic pathway, FIBTEM for the functionality of fibrinogen and APTEM for detection of Heparin [5].

Gathering information about the patient’s coagulation status takes about five minutes [6] for each blood sample. Previous studies investigated the positive effect of ROTEM in relation to specific coagulation factor deficiencies [7,8], the reduction in blood transfusions [9], costs [10,11], and mortality [12] in cardiac surgery [13], liver transplantation [14], obstetrics [15,16], and trauma care [17]. However, more frequent use of ROTEM for early detection and management of coagulopathies would be desirable [10] as the average annual use of ROTEM was never higher than 20% over the period studied (Figure 1).

Figure 1 shows the percentage of trauma patients in whom ROTEM was used as a diagnostic device across all data.

European guidelines state the use of viscoelastic methods (VEM) as additional diagnostic as a grade 1C [18] recommendation. Poor evidence about its benefits [4,19,20] is the main reason for restricting the recommendation. In a recent meta-analytic study by Bugaev et al., the effect of ROTEM is partly described as inconsistent and is only conditionally recommended. Potential benefits in terms of reduced administration of blood products, as well as a reduction in mortality, are described. A reduction in angioembolic/endoscopic/surgical interventions is not reported [21]. Further restrictions are described in a narrative review by Sayce et al. Here, the authors address the fact that ROTEM diagnostics can only provide information about secondary hemostasis. Both the use of antiplatelet drugs and hematologic diseases such as Von Willebrandt syndrome cannot be detected. The influence of alcohol consumption, gender and age of the patients cannot be taken into account in the diagnosis, which is an important limitation in the use of ROTEM [22].

Furthermore, high acquisition costs, other possible diagnostic gaps [23,24], and high personnel expenditure [25] may explain the restrained use of trauma diagnostics.

Besides justified doubts, which are mainly based on a lack of evidence, there are important indications that VEM such as ROTEM can make an important contribution to the understanding of coagulation physiology in trauma patients and improve the care of these patients in the long term [22].

In the present study, we analysed the effect of ROTEM on haemostatic therapy, blood transfusion and mortality rate of severely injured patients in trauma care using data derived from the TraumaRegister DGU^®^ to contribute more evidence on the impact of ROTEM use. This study particularly focuses on the clinical effectiveness of the ROTEM application.

## 2. Methods and Material

### 2.1. Material

The TraumaRegister DGU^®^ of the German Trauma Society (Deutsche Gesellschaft für Unfallchirurgie, DGU) was founded in 1993 [26]. The aim of this multi-centre database is a pseudonymised and standardised documentation of severely injured patients. Data are collected prospectively in consecutive time phases from the site of the accident until discharge from hospital. The infrastructure for documentation, data management, and data analysis is provided by the AUC—Akademie der Unfallchirurgie GmbH (AUC), a company affiliated with the German Trauma Society. The participating hospitals submit their data pseudonymised into a central database via a web-based application. Scientific data analysis is approved according to a peer review procedure established by Sektion NIS. Currently, over 28,000 cases from almost 700 hospitals are yearly entered into the database. The present study is registered as TR-DGU 2016-026N.

In this study, datasets of multiple injured patients documented in the TraumaRegister DGU^®^ (TR-DGU) between 2009 and 2016 were analysed. The inclusion criteria for the analysis were as follows:(1).Only patients admitted to a German hospital;(2).Standard documentation data record of TR-DGU;(3).Worst Abbreviated Injury Scale (AIS) ≥ 3;(4).Documentation of ROTEM use;(5).Primary Admission to a trauma centre (no interhospital transfer);(6).Patients with a risk of bleeding, defined as coagulopathy (PTT ≥ 40, or INR ≥ 1.4 or Quick’s value ≤ 60%) [27] or the need for blood transfusion before ICU admission;(7).Level I or II treating centre;(8).Regional or supra-regional hospitals with >10 trauma cases per year.

Between 2009 and 2016, 238,360 patients were documented in the TR-DGU of which 7461 (3.1%) matched the inclusion criteria. Not matching the eligibility criteria, the following patients were excluded: Of 238,360 patients, 29,795 (12.5%) patients were initially excluded because they were treated outside of Germany. Of these, 115,177 (48.3%) were subtracted because an abbreviated questionnaire was applied. Of the remaining 93,388 patient data, 24,368 (10.2%) mildly injured (max AIS ≤ 2) patients were again excluded. From this, 23,038 (9.67%) Patients for whom no information on ROTEM was provided were then withdrawn. Subsequently, 6603 (2.8%) patients who were transferred to the treating hospital and 1405 (0.6%) patients who were early transferred to another hospital were excluded from this cohort. Of the remaining 37,959 patients, 29,000 (12.2%) data were then removed because they either did not have coagulopathy or did not require transfusions. Furthermore, 132 (0.1%) cases had to be subtracted because they were not treated in at least a level III centre. Patients treated in hospitals where <10 cases per year were documented (1366, 15.5%) were also excluded. 7461 patient data remained and were used for the analysis. All percentages refer to the proportion of the total number of 238,360 documented patients.

Massive transfusion was specified as 10 or more units of packed red blood cells (pRBC) transfused before ICU admission.

### 2.2. Methods

The remaining 7461 patients from 371 hospitals (average 20.1 patients per year) were divided into a ROTEM-using-year-cohort (ROTEM years) and a non-ROTEM-using-year-cohort (Non-ROTEM years) since ROTEM use per hospital was not constant over time. A ROTEM-using year is defined as a year, in which a hospital used ROTEM for ≥20% of patients. Subsumed under this definition, 279 hospital years (corresponding to 5946 patients) were defined as non-ROTEM-using years and 92 hospital years (corresponding to 1515 patients) were defined as ROTEM-using years. Instead of comparing the patients directly, we chose to compare yearly cohorts, in which patients mainly received ROTEM diagnostic or not. On an individual patient level, the use of ROTEM may indicate a more serious situation (selection bias), while in the present investigation we would like to compare treatment strategies. Both groups were compared according to the following criteria:-Hospital mortality;-Risk of death based on the Revised Injury Severity Classification score (RISC II) [28] This score was developed and validated with TR-DGU data and considers age, worst and second worst injury, head injury, pupils, Glasgow Coma Scale (GCS), age and sex, penetrating mechanism, blood pressure, base excess, haemoglobin and prehospital cardiac arrest;-Use of haemostatic therapy: PCC, antifibrinolytics, fibrinogen and tranexamic acid (TXA);-RBC transfusion: units of pRBC;-Probability of massive transfusion (Trauma Associated Severe Haemorrhage (TASH) [29] Score.

Related to these parameters, ROTEM years and Non-ROTEM years were compared. Additionally, ROTEM-year associated data were divided into two groups: one group in which the patients actually received ROTEM diagnostics (ROTEM years with ROTEM, *n* = 790) and one in which they did not receive it but were admitted to a centre in which ROTEM was usually used (ROTEM years without ROTEM, *n* = 725). The division of the groups can be observed in Figure 2.

Figure 2 illustrates the separation of patients into different groups. ROTEM years: a group of patients treated in hospitals in which >20% of patients underwent ROTEM diagnostics. Non-ROTEM years: data were obtained from hospitals where the use of ROTEM was below 20%. Patients in ROTEM years were further divided into ROTEM years with ROTEM and ROTEM years without ROTEM. ROTEM years with ROTEM is the patient collective in which ROTEM was actually used, patients from ROTEM years without ROTEM are those in whom no ROTEM was used for diagnostic purposes, although they were treated in a hospital where ROTEM tends to be used.

By applying certain scores such as the TASH or RISC II score, the comparability was objectified.

Data are presented as mean ± standard deviations (SD) for continuous variables or percentages for categorical variables. Differences between groups were analysed using the Chi-Square Test and the Mann–Whitney U Test. For all statistical analyses, a probability of less than 0.05 was considered to be statistically significant, which is marked with “*” in diagram columns. All data were analysed by using SPSS version 22 (IBM Inc., Armonk, NY, USA).

## 3. Results

During the eight-year period (from 2009 to 2016), data from 7461 patients matched the inclusion criteria. Patients in ROTEM-using years and Non-ROTEM-using years did not differ significantly according to baseline demographics or injury mechanism. Similarly, injury patterns and major clinical treatments, such as intubation rate or CPR rate did not vary. ROTEM was used on 790/1515 patients (53%) of the patients in ROTEM years and on 137/5946 patients (2%) in non-ROTEM years. Details are displayed in Table 1. The application of ROTEM in the period 2009–2016 is shown in Figure 1.

Table 1 compares ROTEM years and Non-ROTEM years regarding their initial conditions (individual patient factors which directly or indirectly influence the probability of survival). The results show similarity between both groups, allowing adequate comparison regarding outcome criteria.

Since the baseline parameters of ROTEM years with ROTEM use and ROTEM years without ROTEM use differ partly significantly, lower comparability is to be generated here than in the comparison of ROTEM years and Non-ROTEM years. Nevertheless, it should be noted, the RISC II score is nearly the same in ROTEM years with ROTEM use and ROTEM years without ROTEM use.

### 3.1. Haemostatic Therapy

The administration of haemostatics regarding prothrombotic (PCC and fibrinogen) and antifibrinolytics (tranexamic acid and other antifibrinolytics) was compared between ROTEM- and non-ROTEM years, furthermore within ROTEM years between patients with and without ROTEM use. The rate of patients who received haemostatic therapy was 56.1% (850/1515) in ROTEM years and 45.7% (2717/5964) in Non-ROTEM years yielding a significant difference between both groups (*p* < 0.001). Haemostatics administered was the highest in ROTEM years with ROTEM use (553/790, 69.5%) and the lowest in ROTEM years without ROTEM use (302/725, 41.7%). The most distinct difference was observed in the prothrombotic class for fibrinogen supplementation which was almost twice as high in ROTEM years with ROTEM use compared with ROTEM years without ROTEM use (54.4% vs. 29.1%, *p* = <0.001). Tranexamic acid (TXA) rates as antifibrinolytic were rather similar across all groups and did not show statistically significant differences whether in comparison between ROTEM- and non-ROTEM years (*p* = 0.14) nor between ROTEM years with and without ROTEM use (*p* = 0.12). Results from all groups can be reviewed in Figure 3.

Figure 3 illustrates the comparison of haemostatic therapy between ROTEM years, Non-ROTEM years, and ROTEM years with and without ROTEM. Differences in general haemostatic therapy, prothrombin concentrate (PCC), antifibrinolytics and fibrinogen were significant in comparison between ROTEM years with ROTEM and ROTEM years without ROTEM.

### 3.2. RBC Transfusion Rates

155/1515 (10.2%) of the patients in ROTEM years and 684/5946 (11.5%) in Non-ROTEM years had received massive RBC transfusions (MT) (*p* = 0.14; Figure 4). In addition, the Trauma-associated Severe Haemorrhage (TASH) scores were compared. The scores were similar in both groups: 14.5% in ROTEM years, 14.6% in Non-ROTEM years (*p* = 0.38). Furthermore, MT rates and TASH scores were compared between ROTEM years with and without ROTEM use. The actual transfusion rate was 14.0% (110/790) in ROTEM years with ROTEM use and 6.1% (44/725) in ROTEM years without ROTEM use (*p* ≤ 0.001). The TASH Score in ROTEM years with ROTEM use was 17.3% and 11.5% in ROTEM years without ROTEM use (*p* < 0.001). The probability of massive transfusion according to the TASH scores was highest in ROTEM years with ROTEM use.

Figure 4 shows the comparison of the TASH score and massive RBC transfusion rates (>10 red blood cell concentrates). Across all groups, the TASH score was higher than the actual massive transfusion rate. Patients who actually received ROTEM diagnostics showed significantly higher massive transfusion rates (14.00%) and TASH scores (17.26%) than in ROTEM years without ROTEM, who had the lowest transfusion rates (6.1%) and the lowest TASH scores (11.47%) across all groups.

### 3.3. Mortality

The probability of death according to RISC II was 30.8% in ROTEM years and 32.3% in Non-ROTEM years (*p* = 0.13; Figure 5). The observed mortality rate was similar in both groups (ROTEM years = 33.2% (503/1515); Non-ROTEM years = 33.0% (1962/5946; *p* = 0.86) and higher than estimated. The difference between the observed versus predicted mortality within one group was +2.4% in ROTEM years and +0.9% in Non-ROTEM years. In ROTEM years with ROTEM use, the average predicted probability to die by RISC II was 30.8% and the observed mortality rate was 30.6% (242/790; difference: −0.2%). An average probability to die of 30.8% and an actual mortality rate of 35.6% (258/725, difference: +4.8%) were achieved in ROTEM years without ROTEM use. Thus, the mortality rate of patients in ROTEM years with ROTEM use was significantly lower than in ROTEM years without ROTEM use (difference: −5.0%, *p* = 0.043) and even the lowest in all groups.

Figure 5 shows the comparison of RISC II scores and actual mortality rates between ROTEM years, Non-ROTEM years, and ROTEM years with and without ROTEM. Mortality rates in ROTEM years and Non-ROTEM years were quite similar, and differences did not show statistical significance (33% vs. 33.2%, *p* = 0.86). ROTEM years with ROTEM show the lowest RISC II score (30.80%) and mortality rate (30.6%) across all groups. It was significantly lower than the mortality rate of ROTEM years without ROTEM (35.6%; *p* = 0.043).

## 4. Discussion

This study examined the impact of ROTEM use compared with conventional coagulation testing on haemostatic therapy, RBC transfusion administration and patient mortality. The approach of comparing hospital-based years rather than direct patient data was used to achieve good comparability.

### 4.1. Haemostatic Therapy

#### 4.1.1. Prothrombotic Agents

Viscoelastic testing offers quick results about potential coagulation disorders including fibrinolysis and hypofibrinogenemia which can be diagnosed with high sensitivity. Especially through the FIBTEM feature which analyses the concentration and function of fibrinogen in the patient’s blood, a fibrinogen-dependent coagulopathy can be diagnosed within minutes [30]. This may explain the significantly more frequent use of fibrinogen in ROTEM years. Similar results were shown in the study by Campbell et al.: There was also a significantly higher administration of fibrinogen in the ROTEM cohort compared with the Conventional Coagulation Testing (CCT) cohort [31]. Other studies have shown that the clot strength parameters after 5 min provide a very reliable indicator of the fibrinogen concentration [32]. This enables a targeted substitution instead of fixed resuscitation protocols [18,33]. Fibrinogen degradation can be observed early after trauma and is considered to be an independent predictor of poor outcomes and increased mortality as a low fibrinogen concentration correlates with a higher risk for ubiquitous microvascular bleeding. Continuous monitoring of fibrinogen concentrations and early substitution of fibrinogen are associated with improved outcomes for the patient [34]. Therefore, targeted fibrinogen substitution plays a key role in haemostatic therapy in trauma care [35].

#### 4.1.2. Antifibrinolytic Agents

Tranexamic acid was the most frequently used agent across all groups without statistical significance. The high administration rate of TXA can potentially be explained by the latest recommendation of the European guidelines: TXA should already be administered in the event of suspicion, without any available evidence, of hyperfibrinolysis [36]. Furthermore, TXA is considered to be cost-effective [36,37] and should be given within 3 h after injury which implies the preclinical administration [38]. The positive effects of TXA on the therapy of coagulation disorders and mortality reduction [39,40,41,42] are well described in the literature. However, studies have shown that tranexamic acid should be used with caution as it can trigger an inflammatory response, which was observed in a perioperative setting [37]. Other side effects reported include thromboembolic events and a higher risk of seizure [43].

Haemostatic interventions do cause adverse events [43]. They can also represent an important cost factor [18]. Further studies should focus on analysing, e.g., the adverse effects of any direct oral anticoagulants which may represent a major challenge in the treatment of trauma haemorrhage and, therefore, are associated with increased mortality of pre-medicalized patients [18].

#### 4.1.3. RBC Transfusion Rates

Regarding the administration of packed red blood cells (pRBCs), the comparison between groups showed no evidence that the use of ROTEM correlates with a reduction in RBC transfusions. Unfortunately, the registry does not document if a defined massive transfusion protocol was activated. Therefore, and due to the retrospective character of this study, the process of decision making by the treating physicians is not transparent. The inability to identify these causalities restricts the evidence of the present analysis. However, even though a negative correlation between ROTEM diagnostic and RBC transfusion rate cannot be observed in our study, an association has been documented in previous studies [44,45]. In this context, it has been described that ROTEM use decreases the transfusion rate of pRBC, FFP and thrombocytes and therefore reduces cost, transfusion-related complications and hospitalisation periods in a wide variety of clinical subjects such as trauma care, intra- and perioperative care and obstetrics [10,46,47]. Cost reduction triggered by decreasing transfusion rates was reported to mount up to 4800 EUR per patient [11]. In a prospective study of patients undergoing correction of thoracolumbar deformity, Guan et al. describe the use of ROTEM enabling a more targeted treatment of coagulopathy. They concluded that patients in non-ROTEM groups required significantly more pRBCs during their hospitalisation than patients whose treatment was guided by ROTEM [48]. In summary, evidence is mounting for ROTEM being associated with a reduction in transfusion rates, even though the present study cannot confirm this statement.

### 4.2. Mortality

The conclusion that the use of ROTEM correlates with a reduced mortality rate could be drawn in a statistically significant range only in comparing the groups ROTEM years with and without ROTEM use. The comparison between ROTEM years and Non-ROTEM years revealed no difference in mortality. In fact, the comparison between ROTEM years and Non-ROTEM years is statistically more valid than the comparison within ROTEM years with and without ROTEM use.

Still, this correlation is supported by perceptions of other studies. Hernandez et al. stated that combining standard laboratory procedures and ROTEM analysis may reduce mortality by 80% compared with standard laboratory procedures alone [12]. A study by Gratz et al. on haemostatic management described that especially patients with craniocerebral trauma benefit from ROTEM use in terms of speed and accuracy of coagulation diagnostics [49]. Additionally, in our study, more than 50% of the patients showed an AIS Head > 3, so one can derive a relation here.

As ROTEM quickly provides information about the patient's coagulation deficiencies [49,50], goal-directed therapy with appropriate and specific haemostatics can be initiated immediately [51,52].

The latest randomised controlled ITACTIC study compared whether patients requiring trauma resuscitation had better outcomes when Major Haemorrhage Protocols (MHP) were supported by Viscoelastic Haemostatic Assays (VHA) instead of CCT. The analysis of 396 patients showed no significant difference comparing the outcome. Nevertheless, an important insight was gained: Treatment of a coagulation disorder proceeded on average 21 min faster with ROTEM use compared with CCT [53].

In the aforementioned study by Gratz et al., the time from sample to result showed a significant difference in ROTEM vs. CCT: 9 min for ROTEM vs. 50 min for CCTs. The detection rate of coagulopathies was also remarkable. Out of 32 patients, ROTEM detected coagulopathy in 21 patients whereas CCT detected abnormal coagulopathy parameters in only 5 patients [49].

ROTEM as a good predictor of mortality was also confirmed by the recent retrospective study of Smith et al.: Here, it could be shown that different changes in the APTEM feature allow reliable predictions on mortality [54]. Fittingly, the observational cohort study by Wang et al. also described ROTEM diagnostic, especially the FIBTEM value, as a good predictor of mortality through reliable and early detection of hyperfibrinolysis [55].

In particular, the presence of fibrinogen deficiency associated with weakened clot strength, which can be detected by the FIBTEM feature, is considered to be a factor that increases both morbidity and mortality [38].

## 5. Limitations

Several limitations of the present analysis have to be noted. As this study is a retrospective analysis, only correlations and no causal relations can be described. Due to the large cohort, results must be interpreted carefully for clinical relevance, regardless of their statistical significance. Furthermore, the lack of a standard timing for the measurement of variables and the lack of the actual chronological order of events have to be emphasised. Although probable, the design of the present analysis does not allow us to conclude that ROTEM use leads to a higher administration rate of haemostatics and, therefore, goal-directed therapy. Additionally, the TR-DGU does not document whether a standardised transfusion protocol was activated which might influence the choice of medication and number of pRBCs transfused. Concludingly, there is no concrete information on how ROTEM results led to specific therapies as a causal relationship. Randomised controlled trials comparing the ROTEM-guided and CCT-guided management may potentially contribute evidence to optimise current trauma guidelines.

## 6. Conclusions

The results of the present study suggest that the use of ROTEM testing may be associated with a significantly higher administration of specific haemostatic interventions—both prothrombotic and antifibrinolytic agents—and RBC transfusions in severely injured patients. Furthermore, we observed a positive correlation between ROTEM testing and a reduction in mortality. Randomised controlled trials are necessary to clarify the use of ROTEM in early trauma care.

## Figures and Tables

**Figure 1 jcm-11-06150-f001:**
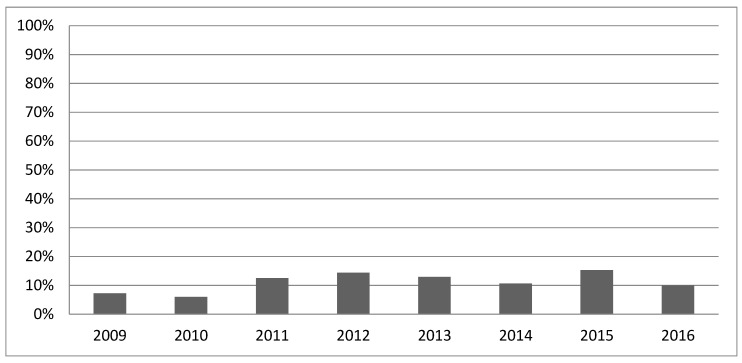
ROTEM application 2009–2016.

**Figure 2 jcm-11-06150-f002:**
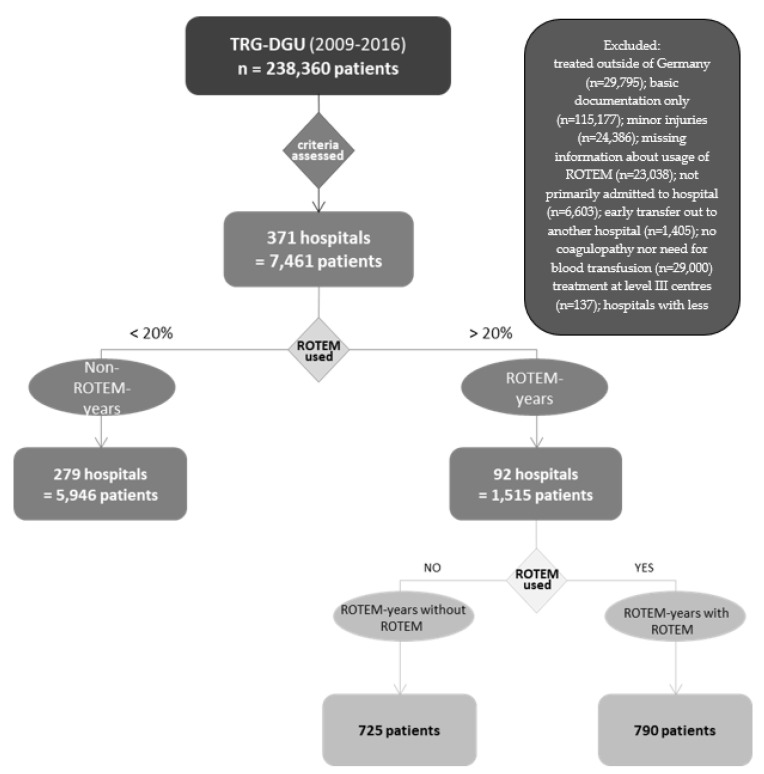
Segmentation of groups.

**Figure 3 jcm-11-06150-f003:**
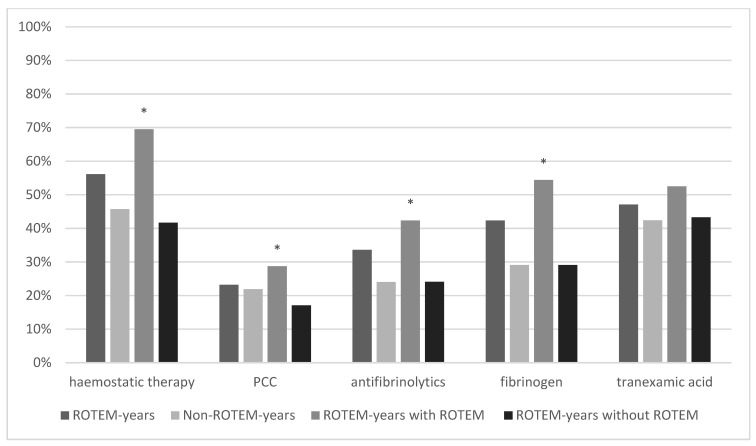
Haemostatic Therapy. * marks statistical significance (*p* ≤ 0001).

**Figure 4 jcm-11-06150-f004:**
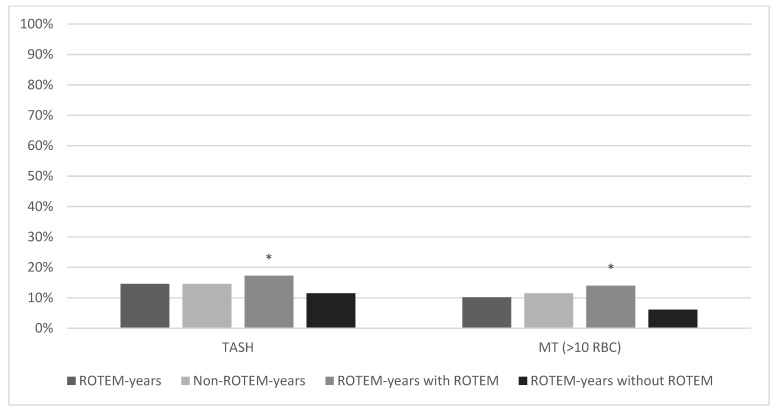
Trauma Associated Severe Hemorrhage score and massive transfusion rate (>10 RBC). * marks statistical significance (*p* ≤ 0.001).

**Figure 5 jcm-11-06150-f005:**
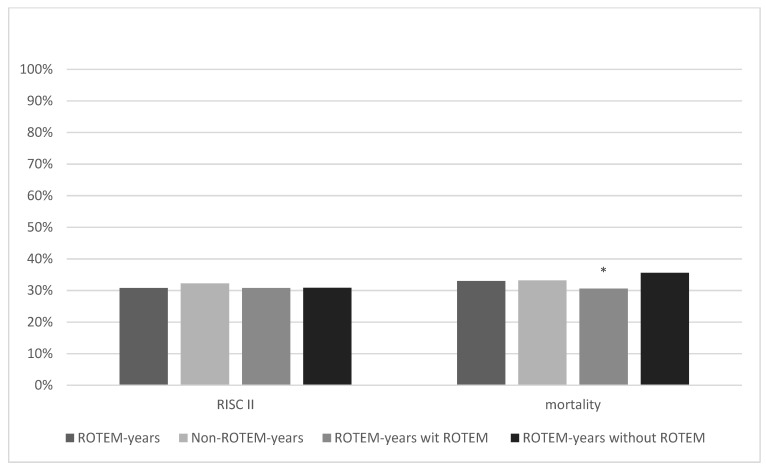
Revised Injury Severity Classification II score and mortality rate. * marks statistical significance (*p* ≤ 0.001).

**Table 1 jcm-11-06150-t001:** Compares ROTEM years and Non-ROTEM years.

	ROTEM Years	Non-ROTEM Years	*p*-Value
Total, *n* (%)	1151 (20.3)	5946 (79.7)	
Male sex, *n* (%)	1078 (71.4)	4065 (68.7)	0.045
GCS ≤ 8, *n* (%)	609 (41.8)	2470 (43.6)	0.236
Blunt trauma, *n* (%)	1375 (93.2)	5417 (94.1)	0.180
AIS Head ≥ 3, *n* (%)	750 (49.5)	3284 (55.2)	<0.001
AIS Thorax ≥ 3, *n* (%)	923 (60.9)	3414 (57.4)	0.014
AIS Abdomen ≥ 3, *n* (%)	366 (24.2)	1377 (23.2)	0.412
AIS Extremities ≥ 3, *n* (%)	725 (47.9)	2675 (45.0)	0.046
SBP ≤ 90 mmHg, *n* (%)	414 (29.9)	1.439 (27.9)	0.149
Intubated, *n* (%)	953 (63.2)	3728 (63.4)	0.881
Resuscitated, *n* (%)	132 (8.7)	510 (8.7)	0.918
MOF, *n* (%)	645 (46.2)	2871 (53.0)	<0.001
average PTT	43.7 (30.1)	44.0 (29.1)	0.062
average INR	1.74 (1.13)	1.75 (1.17)	0.929
average platelet count (in 1000)	192 (84)	189 (89)	0.131

AIS—abbreviated injury scale; GCS—Glasgow Coma Scale; SBP—systolic blood pressure; MOF—multiorgan failure.

## Data Availability

The data presented in this study are available on https://www.auc-online.de/, https://www.traumaregister-dgu.de/ (accessed on 11 July 2022).

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
