# Peer review of "Is ROTEM Diagnostic in Trauma Care Associated with Lower Mortality Rates in Bleeding Patients?—A Retrospective Analysis of 7461 Patients Derived from the TraumaRegister DGU®"

_jcm, 2022, doi:10.3390/jcm11206150_

Round 1
Reviewer 1 Report
Riehl and colleagues present a retrospective review of mortality comparing the use of no ROTEM vs ROTEM. Although the topic is of interest, this paper is very confusing and flawed in methodology. The authors have chosen to compare "years" of ROTEM vs no ROTEM. Although the use of ROTEM at an individual patient level could indeed introduce the confounder of sicker patients being managed with ROTEM (as noted), the simple univariate comparison of years with or without ROTEM is equally if not more substantially flawed. The authors conclude that ROTEM is associated with improved mortality. How can they be sure that ROTEM is not simply a proxy for better or higher quality care? There are numerous hospital level controls for variations in practice that would need to be addressed if this methodology is to be pursued. Simple univariate comparisons were utilized, while a table 1 includes multiple parameters which are different between the groups, yet not controlled for in any subsequent analysis. The level of statistical sophistication is inadequate for such a complex issue with multiple confounders. The study lacks a CONSORT diagram, which makes the exclusion of patients hard to follow. Further, the study stops includes data from 2009-2016 -- it is unclear why the authors chose this range, and resuscitation practices have continued to evolve substantially over this period, which challenges the overall utility of this range. In summary, the methodology of choosing years and univariate comparison only is too premature.
Author Response
Is ROTEM diagnostic in trauma care associated with lower mortality rates in bleeding patients? - A retrospective analysis of 7,461 patients derived from the TraumaRegister DGU®
Katrin Riehl1, Rolf Lefering2, Marc Maegele3, Michael Caspers3 Filippo Migliorini1, Hanno Schenker1, Frank Hildebrand 1 *Matthias Fröhlich3, *Arne Driessen1,4
Dear Reviewers,
Thank you very much for your valuable assessment and scientific comments on the paper submitted "Is ROTEM diagnosis in trauma care is associated with lower mortality rates in bleeding patients? - A retrospective analysis of 7,461 patients derived from the TraumaRegister DGU®".
We would like to address your comments and concerns as follows and hope to clarify outstanding questions and ambiguities.
Riehl and colleagues present a retrospective review of mortality comparing the use of no ROTEM vs ROTEM. Although the topic is of interest, this paper is very confusing and flawed in methodology. The authors have chosen to compare "years" of ROTEM vs no ROTEM. Although the use of ROTEM at an individual patient level could indeed introduce the confounder of sicker patients being managed with ROTEM (as noted), the simple univariate comparison of years with or without ROTEM is equally if not more substantially flawed. The authors conclude that ROTEM is associated with improved mortality. How can they be sure that ROTEM is not simply a proxy for better or higher quality care?
- Thank you very much for your important objection. We cannot be sure about this. It is only a retrospectively observed correlation.
We have tried to make this clear in the limitations.
There are numerous hospital level controls for variations in practice that would need to be addressed if this methodology is to be pursued.
- Thank you very much for your very helpful advice. We did not compare the 279 hospitals without ROTEM with the others that had at least 1 year WITH ROTEM. Presumably, the result is that the "non-ROTEM" hospitals are more often level 2 hospitals, which can lead to a bias, which you have correctly noted.
Simple univariate comparisons were utilized, while a table 1 includes multiple parameters which are different between the groups, yet not controlled for in any subsequent analysis. The level of statistical sophistication is inadequate for such a complex issue with multiple confounders.
The study lacks a CONSORT diagram, which makes the exclusion of patients hard to follow.
- Thank you very much for your very useful comment. We have attempted to present this process in Figure 1 and tried to report exactly how many patients were excluded from our study for which criteria in the text section. Of course, we understand that this is a very redundant section. Ultimately, we find that the information described so correctly.
Further, the study stops includes data from 2009-2016 -- it is unclear why the authors chose this range, and resuscitation practices have continued to evolve substantially over this period, which challenges the overall utility of this range.
- Thank you very much for your valuable contribution. Theoretically, it would be possible to update the period of the collected data. However, in consultation with our statistician, no relevant new findings emerge, so that we also consider the selected period to be representative. 2009 marks the year of the beginning of the records. 2016 was the last year of analysis at the time we started.
In summary, the methodology of choosing years and univariate comparison only is too premature.
- Overall, you are absolutely right with your objections and there are definitely alternative and probably better methods to design the evaluation. In the end, we decided to go this way and try to address the limitations of this methodology in the discussion section. Hopefully, this will allow the study to fit well into the overall context. If you prefer that we fundamentally change the methodology of the study-which is entirely possible-we would indeed have to revise the overall concept and would definitely need much more time than the specified processing period.
Please find the revised version in the appendices.
Thank you very much.
Kind regards,
Katrin Riehl

Reviewer 2 Report
Please see the file attached.

Author Response
Is ROTEM diagnostic in trauma care associated with lower mortality rates in bleeding patients? - A retrospective analysis of 7,461 patients derived from the TraumaRegister DGU®
Katrin Riehl1, Rolf Lefering2, Marc Maegele3, Michael Caspers3 Filippo Migliorini1, Hanno Schenker1, Frank Hildebrand 1 *Matthias Fröhlich3, *Arne Driessen1,4
Dear Reviewers,
Thank you very much for your valuable assessment and scientific comments on the paper submitted "Is ROTEM diagnosis in trauma care is associated with lower mortality rates in bleeding patients? - A retrospective analysis of 7,461 patients derived from the TraumaRegister DGU®".
We would like to address your comments and concerns as follows and hope to clarify outstanding questions and ambiguities.
Abstract: In results, elaborate on ROTEM correlation with specific hemostatic (PCC, FFP, fibrinogen) and antifibrinolytic (antifibrinolytics, TXA) interventions.
- Thank you very much for your comment. Please note the changes we have made in the attached revised version of the paper. There we addressed the issues mentioned.
In conclusions, the conclusion “No relevant differences in transfusion rates could be detected between the groups” contradicts the results obtained. On page 6 (lines: 257- 258), it is indicated that a positive correlation between ROTEM usage and RBC transfusion was detected: “The actual transfusion rate was 14.0% (110/790) in ROTEM 257 years with ROTEM use and 6.1% (44/725) in ROTEM years without ROTEM use. These findings suggest that more transfusions have been prescribed following ROTEM use. Maybe providers felt obligated to do so because ROTEM results were so bad (positive correlation with TASH score). This provider’s decision was informed and based on the data, which is a standard for precision medicine as long as it is in the patient’s best interest. Even if it is not the result authors “hoped” for, they should be open about it and offer an explanation for all their findings – positive and negative.
- Thank you very much for your comment. Please note the changes we have made in the attached revised version of the paper. You are very right with your point. We have tried to implement this by addressing the results holistically.
In conclusions, add your findings on ROTEM use’s impact on hemostatic therapy.
- Thank you very much for your comment. Please note the changes we have made in the attached revised version of the paper. We added all findings on the influence of ROTEM on hemostatic therapy.
Line 70: To improve the clarity of the sentence, after “The effect of ROTEM …” add the word “usage” or “testing”.
- Thank you very much for your comment. Please note the changes we have made in the attached revised version of the paper. We have changed the wording according to your advice.
Line 77: here and on all similar occasions: instead of “blood transfusions” use “RBC transfusions”. It more accurately reflects what the authors were looking at.
- Thank you very much for your comment. Please note the changes we have made in the attached revised version of the paper. We have changed the expressions across the whole text.
Line 79 and other similar occasions: instead of “hemostatic drugs” use “hemostatic therapy” or “hemostatic interventions”.
- Thank you very much for your comment. Please note the changes we have made in the attached revised version of the paper. We have changed the wording according to your advice.
Figures: Fig 3: Data on FFP and ROTEM use are missing. Why is FFP included in methods, but not included in figures and results? Please add the data or explain in the results why they haven’t been included.
- Thank you very much for bringing up this issue which is very true. In fact, we ultimately decided against including the influence of ROTEM on the application of FFP. The fact that this topic can be found in the methods section was indeed a mistake and we are very grateful to you for discovering it.
Fig 3 should be divided into two figures: 1 – presenting results on prothrombotic interventions: hemostatic therapy, PCC, fibrinogen, and FFP; and 2 – antifibrinolytic interventions – antifibrinolytics and TXA.
- Thank you very much for your comment. You are absolutely right with the statement made, actually these two groups should have been treated separately. However, in order to avoid expanding the scope of the paper, we have decided to use a simplified presentation here. Nevertheless, as described under the other comments, we have included a differentiation between the two groups of drugs into the written part.
Figs 3, 4, and 5: add the statistical method used and what “* “ indicates. Clearly state what is “ROTEM-years with ROTEM group” compared to – “ROTEM years without ROTEM”?
- Thank you very much for your comment. Please note the changes we have made in the attached revised version of the paper. We explained the sign and its meaning.
Figure 4: in the graph X-axis group labeling, instead of “>10 RBC” use “MT” or “MT (> 10 RBC)”
- Thank you very much for your comment. Please note the changes we have made in the attached revised version of the paper. We exchanged the expressions as you recommended.
Figures 4 & 5: Do not use abbreviations in figure captions and titles, i.e. TASH and MT.
- Thank you very much for your comment. Please note the changes we have made in the attached revised version of the paper. We have written out the abbreviations accordingly.
Line 205: it is unclear why FFP transfusions are included in the hemostatic therapy group. They should be moved to the blood transfusion category.
- Thank you very much for your comment. Please see the aforementioned comment regarding the FFP issue
Line 206: if authors decide to use RBC transfusions as the only category in blood transfusions (which is against the common practice), please clearly state it in results, abstract, and conclusions. Use “RBC transfusions” instead of “blood transfusions” for clarity.
- Thank you very much for your comment. Please note the changes we have made in the attached revised version of the paper. We exchanged the expressions as you recommended.
In general, it is also not clear why the authors did not include data on transfusion of platelet units. Please offer an explanation in methods and/or limitations.
Consider separating of “Hemostatic therapy” section into two sections – prothrombotic and antifibrinolytic interventions (see suggestions for Fig. 3). Follow this logic in discussion.
- Thank you very much for your comment. Please note the changes we have made in the attached revised version of the paper. We brought in the differentiation in both the results and discussion sections.
Results on FFP are not presented in figures and not discussed in results. Why?
- Thank you very much for your comment. Please see the aforementioned comment regarding the FFP issue
Data on PCC and antifibrinolytics, presented in Fig. 3 suggest positive correlations between the implementation of these interventions along with ROTEM use. Strangely these findings are not addressed in the manuscript narrative. Describe these results in your results and discussion sections.
Not clear why platelet transfusions data are not taken into consideration.
- Thank you very much for bringing up this topic. Unfortunately, we have not collected any data on this. However, it would certainly be an important aspect and should be included in further research.
Line 253: elaborate on what massive transfusion implied, i.e. transfusion of 10 units of packed red blood cells (PRBCs) within a 24 hour period.
- Thank you very much for your comment. Please note the changes we have made in the attached revised version of the paper. We exchanged the expressions as you recommended.
Line 260: should be “massive transfusions”, not “mass transfusion
- Thank you very much for your comment. Please note the changes we have made in the attached revised version of the paper. We exchanged the expressions as you recommended.
Lines 256-261: Positive correlation between ROTEM use and increased RBC transfusions could be associated with the severity of those patients as indicated by the positive correlation between ROTEM use and TESH score. Please discuss this result in the discussion section and properly address in your conclusions. Do not add ambiguous or dismissive statements in conclusions.
- Thank you very much for your comment. Please note the changes we have made in the attached revised version of the paper.
As suggested for results and figures, separately address prothrombotic and antifibrinolytic interventions. Add discussion on PCC, FFP, and antifibrinolytics. Considering the wide audience of the journal, briefly explain what each intervention is for – PCC, TXA etc.
Line 306: “A positive correlation between ROTEM use and haemostatic therapy does not necessarily have a positive impact on patient’s condition”. This statement is not clear and is not supported by the researcher’s data. Also, it contradicts the statement below (Line 326…). Once again, FFP belongs to transfusion of blood products, not to hemostatic therapy. This sentence would sound better when targeted toward the associated risks of transfusion. Transfusions are often the only option for saving a patient’s life – that is a positive impact on the patient’s condition.
- Thank you very much for your comment. Please note the changes we have made in the attached revised version of the paper. We dealt with this important issue and tried to reflect this accordingly.T
Line 320: properly address your findings on the positive correlation of RBC transfusions and TASH with ROTEM use.
- Thank you very much for your comment. Please note the changes we have made in the attached revised version of the paper. We have addressed this subject respectively.
Conclusions: Are rather rudimental. Please clearly state all your findings, including unexpected ones.
- Thank you very much for your comment. Please note the changes we have made in the attached revised version of the paper. We have rewritten the final part in more detail.
Thank you very much.
Kind regards,
Katrin Riehl

Reviewer 3 Report
Review JCM ROTEM 2022
Study: retrospective analyses DGU 2009-2016 data for polytraumapatients ROTEM vs non-ROTEM: outcomes 1) use of hemostatics, 2) rate of blood transfusions, 3) mortality.
Results: 7461 patients, lower mortality in ROTEM, more hemostatic use in ROTEM, no difference in transfusion rates.
Overall: High numbers, clearly/transparantly described methods/results, use of ROTEM years versus non-ROTEM-years should be better argumented with proof of bias when selecting specific patients on statistical basis.
Comments:
Methods:
line 133-157: shorten the explanation on the DGU to information necessary for this study. This extensive report on the founding date etc is distracting.
Line 186. Mass transfusion withholds more than only RBC transfusion.
Line 191-199. Although I understand your rationale, this way of not specifically determining ROTEM use on patient basis also causes a great risk of bias, as you mention > 20% patients received ROTEM, which means 80% or less received non-ROTEM! Selection bias regarding more severely injured patients using ROTEM should be determined statistically and not speculated. Can you deliver these results?
Results
Overall: a lot of percentages within the text clouds the interpretability of the results. In addition, the use of ROTEM years and non-ROTEM-years is a little confusing. Again, I understand the rationale. Is there any way to also present number by patients treated by ROTEM or not (possibly in a supplement). Is there a statistical significant difference in baseline patientcharacteristics as you speculate? Then your reason for the ROTEM versus non-ROTEM years is understandable. Are the results comparable in favour of ROTEM when selected on patient.
Discussion
Elaborate on ROTEM vs non-ROTEM years
Author Response
Is ROTEM diagnostic in trauma care associated with lower mortality rates in bleeding patients? - A retrospective analysis of 7,461 patients derived from the TraumaRegister DGU®
Katrin Riehl1, Rolf Lefering2, Marc Maegele3, Michael Caspers3 Filippo Migliorini1, Hanno Schenker1, Frank Hildebrand 1 *Matthias Fröhlich3, *Arne Driessen1,4
Dear Reviewers,
Thank you very much for your valuable assessment and scientific comments on the paper submitted "Is ROTEM diagnosis in trauma care is associated with lower mortality rates in bleeding patients? - A retrospective analysis of 7,461 patients derived from the TraumaRegister DGU®".
We would like to address your comments and concerns as follows and hope to clarify outstanding questions and ambiguities.
line 133-157: shorten the explanation on the DGU to information necessary for this study. This extensive report on the founding date etc is distracting.
- Thank you very much for your comment. Please note the changes we have made in the attached revised version of the paper.
line 186. Mass transfusion withholds more than only RBC transfusion
.
- Thank you very much for bringing up this issue which is very true. In this paper, however, we have settled on a simplified version of the definition due to missing data from the existing retrospectively analyzed database. Accordingly, several publications aiming the topic of blood management and traumatic induced coagulopathy are using this baseline to define mass transfusion [1–5]
Line 191-199. Although I understand your rationale, this way of not specifically determining ROTEM use on patient basis also causes a great risk of bias, as you mention > 20% patients received ROTEM, which means 80% or less received non-ROTEM! Selection bias regarding more severely injured patients using ROTEM should be determined statistically and not speculated. Can you deliver these results?
- Thank you very much for your commenting on selection bias and the statistical method. Comparing ROTEM group with and without ROTEM use may minimize bias effects. As mentioned in the limitation section of the article, the comparability of the baseline parameters is limited.
Results
Overall: a lot of percentages within the text clouds the interpretability of the results. In addition, the use of ROTEM years and non-ROTEM-years is a little confusing. Again, I understand the rationale. Is there any way to also present number by patients treated by ROTEM or not (possibly in a supplement). Is there a statistically significant difference in baseline patient characteristics as you speculate? Then your reason for the ROTEM versus non-ROTEM years is understandable. Are the results comparable in favour of ROTEM when selected on patient.
- Thank you very much for your comment on the methods of this analysis. Comparisons between groups differ in terms of comparability of baseline parameters and may be indirectly mirrored by the difference in RISCII and TASH score, which are based on the parameters mentioned in table 1
Literature
[1] Yücel N, Lefering R, Maegele M, Vorweg M, Tjardes T, Ruchholtz S, et al. Trauma Associated Severe Hemorrhage (TASH)-score: Probability of mass transfusion as surrogate for life threatening hemorrhage after multiple trauma. J Trauma - Inj Infect Crit Care 2006.
[2] Weber CD, Hildebrand F, Kobbe P, Lefering R, Sellei RM, Pape H-C, et al. Epidemiology of open tibia fractures in a population-based database: update on current risk factors and clinical implications. Eur J Trauma Emerg Surg 2019;45:445–53.
[3] Kirchner T, Lefering R, Sandkamp R, Eberbach H, Schumm K, Schmal H, et al. Thromboembolic complications among multiple injured patients with pelvic injuries: identifying risk factors for possible patient-tailored prophylaxis. World J Emerg Surg 2021;16:42.
[4] Fröhlich M, Mutschler M, Caspers M, Nienaber U, Jäcker V, Driessen A, et al. Trauma-induced coagulopathy upon emergency room arrival: still a significant problem despite increased awareness and management? Eur J Trauma Emerg Surg 2019;45.
[5] Driessen A, Fröhlich M, Schäfer N, Mutschler M, Defosse JM, Brockamp T, et al. Prehospital volume resuscitation - Did evidence defeat the crystalloid dogma? An analysis of the TraumaRegister DGU® 2002–2012. Scand J Trauma Resusc Emerg Med 2016;24:42.
Please find the changes in the attached file.
Thank you very much.
Kind regards,
Katrin Riehl

Reviewer 4 Report
This was a retrospective analysis of registry data in 7461 patients from a trauma registry. Patients in whom ROTEM was used had a lower mortality and high use of fibrinogen, PCC, antifibrinolytics. The results are interesting but as the authors conclude, prospective studies are required.
A few comments:
- Table 1: please include column with p-value to confirm the assertion that there was similarity between groups
- There should be a table comparing baseline parameters of ROTEM users vs non-ROTEM users, and ideally include coagulation parameters in this table (eg APTT, PT, fibrinogen, platelets)
- Paragraph 1 of Results - It was difficult to understand the wording of the second half of the paragraph (beginning from ...Comparing the demographic baseline and injury mechanism between ROTEM years with ROTEM...) Please rephrase so its clearer to understand.
- The discussion I felt was a little too long.
Author Response
Is ROTEM diagnostic in trauma care is associated with lower mortality rates in bleeding patients? - A retrospective analysis of 7,461 patients derived from the TraumaRegister DGU®
Katrin Riehl1, Rolf Lefering2, Marc Maegele3, Michael Caspers3 Filippo Migliorini1, Hanno Schenker1, Frank Hildebrand 1 *Matthias Fröhlich3, *Arne Driessen1,4
Dear Reviewers,
Thank you very much for your valuable assessment and scientific comments on the paper submitted "Is ROTEM diagnosis in trauma care is associated with lower mortality rates in bleeding patients? - A retrospective analysis of 7,461 patients derived from the TraumaRegister DGU®".
We would like to address your comments and concerns as follows and hope to clarify outstanding questions and ambiguities.
This was a retrospective analysis of registry data in 7461 patients from a trauma registry. Patients in whom ROTEM was used had a lower mortality and high use of fibrinogen, PCC, antifibrinolytics. The results are interesting but as the authors conclude, prospective studies are required.
A few comments:
- Table 1: please include column with p-value to confirm the assertion that there was similarity between groups
- Thank you for the valuable suggestion. We have calculated the P values accordingly and added them to the table. The P-values are almost exclusively above the significance level, which should justif the comparability. You can see this in the attached corrected document.
- There should be a table comparing baseline parameters of ROTEM users vs non-ROTEM users, and ideally include coagulation parameters in this table (eg APTT, PT, fibrinogen, platelets)
- Thank you very much for your recommendation. Please find the added data in Table 1.
- Paragraph 1 of Results - It was difficult to understand the wording of the second half of the paragraph (beginning from ...Comparing the demographic baseline and injury mechanism between ROTEM years with ROTEM...) Please rephrase so its clearer to understand.
- Thank you for your important comment. We have tried to reword the mentioned section to generate a better understandability. Please find the highlighted changes in the text section.
- The discussion I felt was a little too long.
- Thank you for your helpful note. We have tried to shorten the discussion section a little.

Round 2
Reviewer 1 Report
The authors have acknowledged my concerns but presented no revisions. It is insufficient to simply highlight limitations when the study contains numerous flawed comparisons. The authors note "Presumably, the result is that the "non-ROTEM" hospitals are more often level 2 hospitals, which can lead to a bias." The notion of comparing Level 1 to Level 2 hospitals with no adjusting characteristics and no controls for additional quality measures or resources is beyond simple citation as a limitation and requires major revision. It is insufficient to draw any conclusion based exclusively on univariate analysis. A robust well controlled analysis is warranted if the authors propose to continue - The limitations of this analysis are too extensive at present for reporting.
Author Response
Dear Reviewers,
Thank you for your renewed evaluation of the paper. We very much regret that the latest response did not improve the analysis.
Regarding your comment on the level I and II hospitals bias, excluding such hospitals from analysis that do not have at least 1 "ROTEM year“ would lead to more consistent results. (Mainly smaller hospitals would be excluded leaving data from hospitals that have done both in order to improve comparability.
Regarding the difference in the level of care, level II hospitals in Germany also provide a very high level of care probably resulting in much smaller difference in care than in other countries.
Ultimately, of course, one could also choose a multivariate approach. Multivariate would have to be calculated at the patient level, not at the hospital level.
This means either the classical approach: all predictors from RISC II plus a few additional factors, the dependent variable would then be mortality.
The second option would be propensity score matching, i.e., we calculate the probability of a patient getting ROTEM and then match 2 cases with the same probability, but one actually got ROTEM and the other did not.
However, both approaches would leave the previous concept comparing treatment strategies.
In addition, more time for changes in content and statistics would be needed correspondingly .
Hopefully, we can agree on a common compromise.
Sincerely,
The authors

Reviewer 2 Report
Accept for publication in a revised form. Thanks.
Author Response
Is ROTEM diagnostic in trauma care is associated with lower mortality rates in bleeding patients? - A retrospective analysis of 7,461 patients derived from the TraumaRegister DGU®
Katrin Riehl1, Rolf Lefering2, Marc Maegele3, Michael Caspers3 Filippo Migliorini1, Hanno Schenker1, Frank Hildebrand 1 *Matthias Fröhlich3, *Arne Driessen1,4
Dear Reviewers,
Thank you very much for your valuable assessment on the paper submitted "Is ROTEM diagnosis in trauma care is associated with lower mortality rates in bleeding patients? - A retrospective analysis of 7,461 patients derived from the TraumaRegister DGU®".
Thank you very much for your help to improve our scientific work and thank you very much for your acceptance for publication.